# ZnO Nanosheet-Coated TiZrPdSiNb Alloy as a Piezoelectric Hybrid Material for Self-Stimulating Orthopedic Implants

**DOI:** 10.3390/biomedicines9040352

**Published:** 2021-03-30

**Authors:** Oriol Careta, Jordina Fornell, Eva Pellicer, Elena Ibañez, Andreu Blanquer, Jaume Esteve, Jordi Sort, Gonzalo Murillo, Carme Nogués

**Affiliations:** 1Departament de Biologia Cel·lular, Fisiologia i Immunologia, Universitat Autònoma de Barcelona, E-08193 Bellaterra (Cerdanyola del Vallès), Spain; oriol.careta@uab.cat (O.C.); elena.ibanez@uab.cat (E.I.); andreublanquer@gmail.com (A.B.); 2Departament de Física, Universitat Autònoma de Barcelona, E-08193 Bellaterra (Cerdanyola del Vallès), Spain; jordina.fornell@uab.cat (J.F.); jordi.sort@uab.cat (J.S.); 3Instituto de Microelectrónica de Barcelona, IMB-CNM (CSIC), C/del Til·lers, Campus UAB, E-08193 Bellaterra (Cerdanyola del Vallès), Spain; Jaume.Esteve@imb-cnm.csic.es; 4Institució Catalana de Recerca i Estudis Avançats (ICREA), Pg. Lluís Companys 23, E-08180 Barcelona, Spain

**Keywords:** TiZrPdSiNb alloy, piezoelectric, nanogenerators, self-stimulating, osteoblast, proliferation, differentiation

## Abstract

A Ti-based alloy (Ti_45_Zr_15_Pd_30_Si_5_Nb_5_) with already proven excellent mechanical and biocompatibility features has been coated with piezoelectric zinc oxide (ZnO) to induce the electrical self-stimulation of cells. ZnO was grown onto the pristine alloy in two different morphologies: a flat dense film and an array of nanosheets. The effect of the combined material on osteoblasts (electrically stimulable cells) was analyzed in terms of proliferation, cell adhesion, expression of differentiation markers and induction of calcium transients. Although both ZnO structures were biocompatible and did not induce inflammatory response, only the array of ZnO nanosheets was able to induce calcium transients, which improved the proliferation of Saos-2 cells and enhanced the expression of some early differentiation expression genes. The usual motion of the cells imposes strain to the ZnO nanosheets, which, in turn, create local electric fields owing to their piezoelectric character. These electric fields cause the opening of calcium voltage gates and boost cell proliferation and early differentiation. Thus, the modification of the Ti_45_Zr_15_Pd_30_Si_5_Nb_5_ surface with an array of ZnO nanosheets endows the alloy with smart characteristics, making it capable of electric self-stimulation.

## 1. Introduction

To be considered as a permanent orthopedic implant, a material must fulfil various criteria. From the mechanical point of view, the material should exhibit high strength, high elastic strain limit and a relatively low Young’s modulus to avoid the so-called stress-shielding effect and the subsequent implant loosening [1,2,3]. In turn, the selected material should exhibit good biocompatibility and biostability (i.e., corrosion and hydrolysis resistance), osseointegration, high wear resistance and high bioinertness (i.e., non-toxic, non-allergenic and non-carcinogenic) [3,4,5]. Among the studied materials, Ti and its alloys (Ti_6_Al_4_V, Ti_40_Nb) have been established as the most suitable biomaterials for permanent implants in certain applications (i.e., bone and joint replacements) because they exhibit superior mechanical properties, biocompatibility and corrosion resistance [6,7], when compared with other metallic implants, such as stainless steel or Co–Cr alloys [3,4,8].

Nevertheless, these alloys face some problems, such as the presence of non-desirable elements (e.g., Al and V in Ti_6_Al_4_V alloys [9,10]) or the mismatch between their Young’s modulus (i.e., ~110 for Ti-6Al-4V [7] and 40–80 for Ti-Nb alloys [11]) and that of the bone (10–30 GPa) [12]. In view of this, ongoing investigations on Ti-based implants are focused on finding new compositions and designs to improve their performance [4,13,14,15]. One example is our previously designed Ti_45_Zr_15_Pd_30_Si_5_Nb_5_ alloy, with high hardness, a competitive Young’s modulus value, and excellent biocompatibility [4].

On the other hand, it is possible to improve the performance and durability of the bioimplants, as well as to modulate the behavior of cells in contact with them, by modifying their surface. Surface modification (i.e., film deposition, laser etching, acid and alkali treatments or anodic oxidation) is an appropriate strategy to modify the surface-dominant properties of the implant without compromising those of the bulk material [3,15]. Surface treatments can be employed to improve strength and wear resistance, avoid ion release, or encourage bioactivity between the bone and other tissues [3,13]. Coating Ti and Ti alloy implants with bioactive ceramic coatings, such as hydroxyapatite (Ca_10_(PO_4_)_6_(OH)_2_), which is the principal constituent of hard tissues, is the most explored method to improve bioactivity [13,16,17,18]. Bioceramic coatings encourage the growth of new bone cells, inhibit the release of metal ions into the surrounding bone tissues, and may promote an earlier osteoid formation. However, the main drawbacks of the hydroxyapatite coatings are the low bond strength and the high residual stress, which may cause its detachment from the implant during long-term service in the human body. Type I collagen coatings have also been considered; some studies have reported their ability to modulate various aspects of cell behavior, such as cell attachment, proliferation, and differentiation on Ti-based implants [12,19,20,21].

To further increase the performance of bioactive implants, stimulus-responsive biomaterials (i.e., the so-called smart biomaterials) are currently the focus of research because they can also actively participate in the restoration of damaged tissues in response to stimuli (i.e., pH, temperature, electric field, etc.) from their biological environment [3]. To date, most of the reported stimulus-responsive scaffold-based bone implants are made of biodegradable polymers [22,23,24] and ceramic materials [25,26], but the integration of stimuli-responsive coatings deposited onto metallic scaffolds with excellent mechanical performance remains a challenge.

Within this framework, piezoelectric materials are particularly appealing because they can stimulate electrically excitable cells (e.g., cardiomyocytes, skeletal myotubes, osteoblasts and neural cells) and modulate cell activity as a result of a mechanical stimulus using a non-invasive technique. Due to the piezoelectric effect, these materials can create a charge separation, and a subsequent electric potential, when they are mechanically stressed. To trigger the electrical stimulation, various sources of mechanical actuation, including vibration plates, sounds and ultrasounds, can be employed [27,28,29]. In fact, piezoelectric nanostructures (i.e., nanoparticles, nanobelts, nanotubes, nanofibers, nanosheets or nanorods) made of boron nitride, barium titanate, ZnO or polyvinylidene fluoride (PVDF) are attracting strong interest as neural, muscular, or bone cell stimulators [30,31,32,33,34,35,36,37].

In particular, nanostructured ZnO is widely explored in energy harvesting applications for its ability to generate voltage when mechanically stressed [38], although its use for the electrical stimulation of living cells is much less explored probably because of the contrasting reports about its biosafety [35,39,40]. Nonetheless, Ciofani et al. [35] demonstrated the suitability of ZnO nanowire arrays in sustaining cellular functions and their potential in applications of tissue engineering and invasive sensing or stimulation. However, no detailed studies to evaluate the effect of stimulation due to the piezoelectric effect have been carried out. In an attempt to elucidate the biocompatibility and the electromechanical ZnO–cell interaction, we previously cultured macrophages and electrically excitable osteoblast-like cells on top of two-dimensional (2D) ZnO nanosheets grown on an AlN-coated Si substrate [41]. During their growth, the cells induced a local strain to the ZnO nanosheets, and due to their piezoelectric properties, an electric potential difference was locally generated in the plasma membrane. These ZnO–cell interactions stimulated the motility of macrophages and triggered the opening of ion channels in the osteoblast-like cells, inducing intracellular calcium transients. These results opened the door to promising applications of piezoelectric ZnO nanosheets in osteogenesis and bone regeneration. However, silicon is not suited for orthopedic applications.

In the present work, a Ti_45_Zr_15_Pd_30_Si_5_Nb_5_ alloy, previously designed by the authors and presenting high hardness, a competitive Young’s modulus value, and excellent biocompatibility [4], was coated with ZnO with the aim of endowing the hybrid material with a new functionality; namely, the capability to stimulate cells without the need to apply an external electric field. Two different 2D ZnO structures were tested: a flat dense ZnO-sputtered film, and an array of ZnO nanosheets. The effect of the combined material on osteoblasts (electrically stimulable cells) was analyzed in terms of proliferation, cell adhesion, expression of differentiation markers and induction of calcium transients.

## 2. Materials and Methods

### 2.1. Material Fabrication

A master alloy with nominal composition Ti_45_Zr_15_Pd_30_Si_5_Nb_5_ (in at.%) was prepared by arc melting a mixture of the highly pure elements (>99.99% wt.%) under a Ti-gettered Ar atmosphere. Rods of 5 mm in diameter were obtained from the melt by suction casting into a Cu mold. Further details on sample fabrication and microstructural characterization can be found elsewhere [4]. The as-cast, pristine alloy was cut in disks of about 1 mm thick and one-side polished with 4000 grid size SiC paper as a final step for posterior ZnO deposition. The polished surface of the metallic alloy (referred to as “pristine” alloy) was further modified with ZnO, which was deposited in two different ways: (i) covering the alloy with a thin layer of ZnO (referred to as “ZnO thin film”); and (ii) covering the alloy with an array of 2D ZnO nanostructures (referred to as “ZnO nanosheets”) (Figure 1).

The ZnO thin film was obtained by sputtering a Zn target at 100 W DC for 10 min under a reactive O_2_ atmosphere. The ZnO nanosheets were grown by hydrothermal synthesis. For this, the Ti_45_Zr_15_Pd_30_Si_5_Nb_5_ alloy substrates were placed in a supporting 4-inch silicon wafer to deposit a layer of 1 µm of SiO_2_ by plasma-enhanced chemical vapor deposition as a buffer layer, to avoid the cracking of the posterior layers due to thermal expansion gradients. These processes were performed in a class 100–10,000 cleanroom, with controlled conditions of temperature, pressure, and humidity. Then, a thin-film layer of 100 nm AlN was deposited by using radio-frequency sputtering on top of the SiO_2_ layer. An aqueous solution containing hexamethylenetetramine and Zn(NO_3_)_2_ was poured in a Pyrex container [42,43]. The supporting wafer was placed in the container fixed by four polyether ether ketone screws, with the AlN layer facing down. The container was hermetically closed and introduced in an oven at 80 °C for 9 h. After the growth was completed, the supported wafer was collected and rinsed in deionized water and ethanol and left to dry at room temperature (RT). Finally, the coated alloy substrates were carefully released from the wafer. The growth parameters (i.e., 80 °C for 9 h and the aqueous concentration) were optimized in order to improve the activation effect in osteoblast-like cells.

### 2.2. Structural Characterization

The morphology of the pristine alloy, the ZnO thin film and the ZnO nanosheets deposited on top was studied by scanning electron microscopy (SEM) on Zeiss Auriga and Merlin microscopes.

The composition of the samples was determined by energy-dispersive X-ray spectroscopy (EDS/EDX). EDX measurements of the base alloy were carried out at 15 kV, whereas those on the ZnO structures were obtained at lower voltages (2–5 kV) in order to restrict the penetration of the X-rays to the utmost ZnO. Grazing incidence X-ray diffraction (GIXRD) analyses were conducted on a Malvern-PANalytical X’Pert Pro MRD diffractometer using CuKα radiation for phase analysis of the samples in a 2θ range from 30° to 80°.

### 2.3. Cell Culture

Human osteosarcoma Saos-2 cells (ATCC HTB-85) were cultured in Dulbecco’s modified Eagle’s medium (DMEM) (Gibco, ThermoFisher Scientific, Waltham, MA, USA) supplemented with 10% fetal bovine serum (FBS; Gibco), under standard conditions (37 °C, 5% CO_2_). Saos-2 cell line derives from osteosarcoma, and even though it could show some minor disparities in terms of cellular behavior when compared with primary human osteoblasts, Saos-2 cells have been widely used to test the biocompatibility of newly developed materials for a wide range of biomedical applications, being considered as a representative model of multiple osteoblast responses [44]. The samples (pristine alloy, used as controls in all experiments, ZnO thin film, and ZnO nanosheets) were sterilized with absolute ethanol for 30 min and individually introduced into a 4- or a 24-well plate. Then, different numbers of cells, depending on the experiment, were seeded into each well.

THP-1 monocyte cells were used to analyze the immunological response to alloy samples. Monocytes were grown in RPMI 1640 medium (Gibco) supplemented with 25% FBS under standard conditions. To differentiate monocytes into macrophages, 400,000 THP-1 cells were seeded into 24-well plates and treated with 0.16 µM phorbol-12-myristate-13-acetate (Sigma-Aldrich, Saint Louis, MO, USA) for 72 h. Then, cells were washed and incubated in fresh medium for 24 h before carrying out the experiments.

### 2.4. Quantification of Cell Proliferation

Osteoblast proliferation was determined using Alamar Blue cell viability reagent (Thermo Fisher Scientific) at days 1, 3 and 7. Briefly, 250,000 cells were seeded into each well of a 24-well plate containing each sample (pristine alloy, ZnO thin film and ZnO nanosheets). After 24 h, samples with adhered cells were moved to a new well to discard cells growing outside the surface of the alloys. Fresh medium with 10% Alamar Blue was added, and cells were incubated for 4 h in the dark and standard conditions. Then, the supernatant was collected, and its fluorescence was measured at 585 nm wavelength after excitation at 560 nm on a Varian Cary Eclipse Fluorimeter (Agilent Technologies, Santa Clara, CA, USA). Fresh medium was added to the cultures and the assay was repeated after 3 and 7 days. Experiments were performed in triplicate.

### 2.5. Quantification of Inflammatory Cytokines Secretion

To conduct these experiments, THP-1 cells differentiated into macrophages were used. A previously sterilized alloy sample was placed on a 6.5 mm Transwell^®^ with a 0.4 µm Pore Polyester Membrane Insert (Corning, Corning, NY, USA) placed over the macrophage cultures (cell culture insert system) and maintained in close contact with the cells for 24 h to analyze the secretion of inflammatory cytokines. As a positive control, 1 µg/mL of lipopolysaccharide (LPS) (Sigma-Aldrich) was added to the macrophages culture. As a negative control, macrophages were cultured in the absence of the alloys or LPS. After 24 h, supernatants were collected and used to quantify cytokine secretion. Inflammatory cytokines TNF-α, IL-1β and IL-6 were evaluated by flow cytometry using cytometric bead array (CBA) (Becton–Dickinson, East Rutherford, NJ, USA). Cytokine concentrations in the supernatant were analyzed according to the manufacturer’s protocol. Negative control was considered as basal secretion level with a 0 value. Experiments were performed in triplicate.

### 2.6. Cell Adhesion Analysis

Cell adhesion was determined through the analysis of focal contacts by actin filaments and vinculin detection. In these studies, 100,000 cells were seeded onto the different samples and, after 72 h, alloys were washed twice in PBS and cells fixed in 4% paraformaldehyde in PBS for 15 min at RT. After washing twice in PBS, cells were permeabilized with 0.1% Triton X-100 (Sigma-Aldrich) in PBS for 15 min and blocked for 25 min with 1% bovine serum albumin (BSA) (Sigma-Aldrich), 0.5% Tween 20 (Sigma-Aldrich) in PBS at RT. Samples were then incubated with a mouse anti-vinculin primary antibody (Millipore, MAB3574) at 2 µg/mL overnight at 4 °C and washed with 1% BSA-0.5% Tween 20 in PBS. Then, samples were incubated with a mixture of Alexa fluor 594-conjugated phalloidin (Invitrogen, ThermoFisher Scientific, Waltham, MA, USA), Alexa fluor 488 chicken anti-mouse IgG (Invitrogen), and Hoechst 33,258 (Sigma-Aldrich) for 60 min in the dark at RT. Finally, cells were washed in PBS, air-dried, and mounted on specific bottom glass dishes (MatTek, Ashland, MA, USA) using ProLong Antifade mounting solution (Life Technologies, Carlsbad, CA, USA). Immunofluorescence evaluation was performed in a confocal laser scanning microscope (CLSM) (Olympus, Shinjuku, Tokyo, Japan).

### 2.7. Cell Morphology Analysis

A total of 250,000 cells were seeded into each well of a 24-well plate containing each type of sample and cultured for 7 days. Then, cells were processed to be analyzed by SEM. Briefly, cells were washed in cold phosphate buffered saline (PBS), fixed in 4% paraformaldehyde in PBS for 15 min at RT, and washed again in PBS. Cell dehydration was performed in a series of increasing ethanol concentrations (50, 70, 90 and twice 100%) for 8 min each. Finally, samples were dried using hexamethyldisilazane (Electron Microscopy Sciences, Hatfield, PA, USA) for 15 min. Samples were mounted on special stubs and analyzed using an SEM Merlin (Zeiss, Oberkochen, Germany) in order to observe cell morphology.

### 2.8. Intracellular Calcium Measurement

Time-lapse CLSM (Leica SP5) (Leica, Wetzlar, Germany) was used to measure the intracellular calcium dynamics over time. Briefly, 100,000 Saos-2 cells were seeded on a 4-well plate containing an alloy sample. Cells were incubated for 24 h in standard conditions, and then were loaded with 2 μM Fluo-4 AM (Life Technologies) in serum-free DMEM for 30 min in the dark at RT. Samples were washed with serum-free DMEM and then placed upside-down into MatTek dishes with fresh medium without phenol red. Images of osteoblasts were captured in time-lapse CLSM every 1 s for 30 min. Changes in fluorescence intensity during the time of monitoring were processed using Image J software.

A MATLAB code was developed to automatically detect ββ^+^ influx in cells, using the time-lapse videos recorded in the CLSM as the data source. This code uses several image enhancement routines and perimeter detection algorithms to identify all the cultured cells. The centroid, pixel coordinates, and the area of each cell larger than 30 pixels are stored in a data structure to be processed. Then, for each particular cell, the mean relative intensity over time was calculated and used as the input of an ad hoc peak detector to evaluate whether the cell was electrically activated by the associated ZnO piezoelectric effect or not.

### 2.9. Quantitative Real-Time PCR

The expression of osteogenic marker genes encoding alkaline phosphatase (ALPL), osteocalcin (BGLAP), type I collagen (COL1), bone sialoprotein (IBSP), osteonectin (SPARC) and osteopontin (SPP1) was analyzed by real-time quantitative polymerase chain reaction (qPCR). For gene expression analysis, 100,000 cells were seeded onto samples and, at 7, 14 and 21 days, total RNA was extracted from the cell cultures using the Maxwell RSC simplyRNA tissue kit (Promega, Madison, WI, USA) according to the manufacturer’s protocol. RNA concentration and purity were determined using a Nanodrop spectrophotometer (Nanodrop 1000, Thermo Scientific, ThermoFisher Scientific). Reverse transcription was performed with 500 ng total RNA using the iScript cDNA synthesis kit (BioRad, Hercules, CA, USA), according to the manufacturer’s instructions. The mRNA levels were assayed in triplicate in CFX384 arrays (BioRad) using 5 µL of iTaq Universal SYBR Green Supermix (BioRad), 0.5 µL of PrimePCR Assays (BioRad) and 20 ng cDNA in a total volume of 10 µL. The PCR amplification was performed as follows: initial heating at 95 °C for 3 min, followed by 40 cycles at 95 °C for 10 s, 60 °C for 30 s, and a final melt curve from 65 °C to 95 °C, in 0.5 °C increment each 5 s in a C1000 Touch Thermal Cycler (BioRad). Expression values were obtained from cycle quantification (Cq) values determined with the BioRad CFX Maestro™ Software. The target gene levels are expressed as a relative value: the ratio of the target gene expression to that of the reference TATA-box binding protein (TBP) and hypoxantine phosphoribosyltranferase (HPRT1) genes. The relative gene expression was calculated as 2-ΔCq. Validated PrimePCR SYBR Green Assays (BioRad) for ALPL (qHsaCID0010031), BGLAP (qHsaCED0038437), COL1 (qHsaCED0043248), IBSP (qHsaCED0002933), SPARC (qHsaCID0010332), SPP1 (qHsaCID0012060), TBP (qHsaCID0007122) and HPRT1 (qHsaCID0016375) were used.

### 2.10. Statistical Analysis

All quantitative data were analyzed with GraphPad Prism 6 (GraphPad Software Inc., San Diego, CA, USA) and presented as the mean ± standard deviation. Statistical comparisons were performed using one-way analysis of variance (ANOVA) with the Tukey–Kramer multiple comparison test for cell proliferation assays. Multiple comparison procedures were performed with ANOVA using the Student–Newman–Keuls method for gene expression values. A value of *p* < 0.05 was considered to be significant. Significance is represented in the figures using an alphabetical superscript system on top of the columns. Values with different alphabetical superscripts are significantly different, whereas values with the same alphabetical superscripts do not significantly differ.

## 3. Results

### 3.1. Morphology, Composition and Structure of the Samples

The morphology of the pristine Ti_45_Zr_15_Pd_30_Si_5_Nb_5_ base alloy and the ZnO materials deposited on top (Figure 1), i.e., the sputtered thin film and the hydrothermally synthesized nanosheets, can be seen in the SEM images of Figure 2a–c. The composite-like microstructure of the base alloy (the eutectic lamellae in particular) is visible in Figure 2a. The sputtered ZnO thin film was featureless, likely due to is reduced thickness (around 30 nm) and very low surface roughness (Figure 2b). Meanwhile, the surface of the alloy covered by ZnO nanosheets was much rougher, due to the lamellar structure produced by the high aspect ratio and smooth ZnO crystalline array showing a hexagonal sheet morphology, with an average thickness of 25 ± 8 nm (Figure 2c). The constituting elements of the pristine alloy are all present in its corresponding EDX pattern (Figure 2d), with an atomic ratio close to the nominal one (i.e., Ti_42_Zr_17_Pd_31_Si_6_Nb_4_). The EDX pattern of the ZnO thin film (which was acquired at a low voltage to maximize the response from the ZnO layer) shows the signals of Zn and O (Figure 2e). Figure 2f depicts the GIXRD patterns of the pristine alloy and the ZnO nanosheets. The most intense peaks for the base alloy belong to the cubic β-Ti phase (Im3m), although β-Ti reflections partially overlap with α-Ti, as previously demonstrated [4]. On the other hand, the GIXRD pattern of the ZnO nanosheets grown onto Ti_45_Zr_15_Pd_30_Si_5_Nb_5_ shows reflections corresponding to the metallic alloy beneath and new reflections matching the hexagonal phase of ZnO (wurtzite). Note that the ZnO-sputtered thin film did not diffract, probably because of its small thickness.

### 3.2. Cell Proliferation

The proliferation of Saos-2 cells grown on the different samples (pristine, ZnO thin film and ZnO nanosheets) was quantified after one, three and seven days of seeding. Results were normalized with respect to the viability at day 1 and compared among materials at each time-point. As shown in Figure 3, the number of cells growing on the pristine alloy and on the ZnO thin film were similar at all time-points. By contrast, the number of cells growing on ZnO nanosheets at day 7 was significantly higher than on the other two surfaces.

### 3.3. Induction of Inflammatory Cytokine Secretion

To find out whether the elements present in the metallic alloy or the different ZnO coatings activated the secretion of inflammatory cytokines, macrophages were cultured in the presence of the three samples using a cell culture insert system. CBA results showed that the presence of any of the three materials did not activate the secretion of TNF-α, IL-6, or IL-1β in macrophages (Figure 4). However, when macrophages were exposed to LPS (positive control), a significant increase in the concentration of the three cytokines was detected in the culture medium.

### 3.4. Cell Morphology and Adhesion Analysis

Immunofluorescence analysis of vinculin and stress fibers (actin) showed that osteoblasts were completely adhered to the sample surface after three days of culture (Figure 5) and were able to establish focal contacts with all surfaces. Stress fibers were well defined, some of them crossing the totality of the cell and some of them ending in a focal contact. Stress fibers were found either in a parallel orientation or without a defined orientation.

SEM analysis of osteoblasts grown on the different samples showed that cells were randomly distributed on their surfaces after seven days of culture (Figure 6). In all cases, cells presented a flattened polygonal morphology with cytoplasmic extensions in different directions. Moreover, cells established contacts with nearby cells through thin plasma membrane projections, and most of the cells presented several nucleoli in their nucleus. Cells growing on ZnO nanosheets emitted long membrane projections that anchored directly on the nanosheets (Figure 6f).

### 3.5. Induction of Intracellular Calcium Transients

The analysis of intracellular calcium transients induced by the three types of samples can be seen in Figure 7. A low percentage of Saos-2 cells grown on the pristine alloy (8.00% ± 4.58%) and the ZnO thin film (5.33% ± 3.05%) experienced transient increases in Ca^2+^ concentration. By contrast, 46.50% ± 4.95% of Saos-2 cells grown on the ZnO nanosheets were activated, presenting intracellular calcium transients.

### 3.6. Expression of Osteogenic Markers

Finally, we analyzed the expression of six specific osteogenic marker genes encoding alkaline phosphatase (ALPL), osteocalcin (BGLAP), type I collagen (COL1), bone sialoprotein (IBSP), osteonectin (SPARC), and osteopontin (SPP1) after 7, 14 and 21 days in culture on the three samples in standard conditions (Figure 8). In general, marker gene expression was similar or significantly higher on cells grown on ZnO nanosheets when comparing all three materials at 7, 14 or 21 days of culture. The only exceptions were IBSP at day 7, of which the expression was significantly lower on ZnO thin film; COL1 and IBSP at 14 days, of which the expression was significantly lower in cells grown on ZnO nanosheets; and SPARC at day 21, of which the expression was significantly higher in cells grown on ZnO thin film, even though expression in cells grown on ZnO nanosheets was also higher than in cells grown on pristine alloys.

## 4. Discussion

Nanostructured Ti_45_Zr_15_Pd_30_Si_5_Nb_5_ alloy with predominant β-type phase was previously synthesized and characterized by our group [4]. It showed a good combination of mechanical properties, with reduced Young’s modulus around 85 GPa and hardness around 10.4 GPa. Indeed, the addition of 5 at.% Nb to the TiZrPdSi alloy was proven to decrease the Young’s modulus from 117 GPa to 85 GPa, thus making it closer to that of the bone. Importantly, we also demonstrated that the addition of Nb to the alloy enhanced osteoblast-like cell (Saos-2) adhesion and proliferation, and that cells grown onto the TiZrPdSiNb alloy presented higher levels of some late osteogenic markers during the first week in culture [45]. On the other hand, in a separate work, we showed that the interaction of living cells with piezoelectric ZnO nanogenerators induces a local electric field that self-stimulates and modulates cell activity, without the need of an additional chemical or physical external stimulation [41].

In the present study, we combined the characteristics of both materials to obtain a hybrid Ti_42_Zr_17_Pd_31_Si_6_Nb_4_ alloy covered with ZnO nanosheets, capable of acting as an implantable orthopedic material with cell self-stimulation characteristics while, in turn, improving cell proliferation and differentiation. To analyze the effect on cell behavior of the ZnO counterpart as a piezoelectric material, we coated some samples of pristine alloy with a flat, thin, ZnO layer and, separately, others with an array of ZnO nanosheets. We evaluated the self-stimulation activity of cells grown on the hybrid material using Saos-2, because they present voltage-gated calcium channels and stretch-activated cation channels in their membranes [46], and the opening of these channels can be triggered by electric fields. It is known that the external electrical stimulation of osteoblasts increases their proliferation and differentiation [47].

First, biocompatibility was analyzed in terms of cell proliferation on days 3 and 7. None of the ZnO coatings caused a decrease in Saos-2 viability, confirming that ZnO is not cytotoxic. The techniques used to coat the alloy (sputtering and hydrothermal growth) had no negative effect on cell viability (e.g., they did not leave remains of toxic solvents) and the layers remained attached to the alloy during the entire duration of the experiments. Cell proliferation was similar between ZnO thin film and pristine alloy samples at all time points analyzed but, interestingly, ZnO nanosheets increased Saos-2 proliferation on day 7 compared with the other two types of samples. The biocompatibility of the pristine alloy and the ZnO nanosheets had already been determined in previous studies by our group [41,45], but the effect of the flat ZnO layer and the procedures used to attach the ZnO materials to the pristine alloy on cell proliferation had not been investigated. Although ZnO is generally recognized as a safe material by the Food and Drug Administration (FDA) of the United States, controversial results have been reported on its cytotoxicity. It has been described that ZnO nanorods inhibit cell adhesion [48], although ZnO nanoflowers improved cell proliferation when compared with ZnO films [49]. All these results suggest that cytotoxicity depends on the specific shape of ZnO. According to our results, both the ZnO thin film and the nanosheets array are not cytotoxic, and the latter also enhances osteoblast proliferation.

Secondly, and related to biocompatibility, the possibility that these materials could trigger an inflammatory response was analyzed by quantifying the secretion of three cytokines involved in inflammation. None of the materials tested increased the release of inflammatory cytokines under study compared with the pristine alloy. Even though biomaterials are generally non-toxic, they can mediate adverse reactions such as inflammation, and it is important to discard this problem because inflammation may end in implant failure [50,51].

Once the safeness of the hybrid materials was demonstrated, we analyzed the adhesion and morphology of osteoblasts grown on them. It is well known that surface topography can modulate cell adhesion, morphology and cell–material interactions [52,53]. Cell adhesion and spreading are important to prevent aseptic loosening of the implants caused by fibroblast layer attachment [50]. No differences in adhesion or morphology were found among the three samples, even though the topography of the ZnO nanosheets and ZnO thin film or pristine alloy was different. Actin stress fiber distribution and focal contacts were also similar among cells grown on the three materials. It has been described that small differences in the order of nanometers could improve cell attachment and, in turn, proliferation, because osteoblasts need to adhere to proliferate and differentiate [52,53]. As mentioned before, the ZnO nanosheets enhance osteoblast proliferation although, apparently, no differences in adhesion were observed.

The next step was to test whether the samples with the ZnO layer presented piezoelectric properties. A piezoelectric material is a material that generates an electric field when mechanically deformed and vice versa, i.e., it deforms when an electric field is applied. In a previous study [41], we had shown that the typical movements of cells in culture are sufficient to cause small deformations (strain) to the partially clamped ZnO nanostructures, thereby inducing small local electric fields. It is also known that osteoblasts exposed to an electrical field can exhibit changes in their cytosolic calcium concentration [46,54] and, because Ca^2+^ is a secondary messenger involved in numerous transduction pathways [47], these Ca^2+^ transients can enhance proliferation and/or differentiation. Cell adhesion was excellent on all three materials; therefore, one possible explanation for the higher cell proliferation on the ZnO nanosheets compared to the ZnO thin film and pristine alloy could be the existence of an electrical stimulation due to the piezoelectric effect present in the ZnO nanosheets, arising from cell-induced strains. Note that such strains cannot develop in the sputtered ZnO thin film because of the full mechanical clamping with the metallic Ti_45_Zr_15_Pd_30_Si_5_Nb_5_ base alloy, which is rigid. Such clamping is much more limited in the case of ZnO nanosheets due to the reduced interface contact area between the sheets and the base alloy, obtaining nanostructures capable of bending. Hence, our results suggest that only metallic alloys whose surface is covered with 2D nanostructured ZnO (i.e., ZnO nanosheets), as shown here, are able to induce a local electric field large enough to stimulate the cells and alter their activity, allowing the opening of the calcium channels and the influx of calcium from the external medium.

Finally, to study the effect of all materials in osteoblast differentiation, the expression of genes related to osteoblast differentiation were analyzed. The differentiation and maturation of osteoblasts are necessary for bone regeneration, and they play an important role in implant osseointegration. Six different osteogenic genes were studied. Of them, three were related to extracellular matrix synthesis (early expression genes) and the other three were related to extracellular matrix mineralization (late expression genes). However, these two categories are not strictly determined, because the expression of osteogenic genes is a dynamic process influenced by several factors. COL1, ALPL and IBSP genes are considered early markers, although their expression can be moderate or high during all differentiation processes, whereas BGLAP, SPARC, and SSP1 are considered late markers [55]. Cells grown on ZnO nanosheets presented similar or higher levels of all genes analyzed (except for IBSP at 14 days) compared with cells grown on pristine alloy. In addition, cells grown on ZnO nanosheets presented similar or higher levels of the genes analyzed compared with cells grown on the ZnO thin film, except for COL1 and IBSP on day 14 and SPARC on day 21. Overall, these results suggest that ZnO nanosheets enhance the expression of some osteoblast differentiation markers, mainly early expression genes. As previously discussed, the increase in differentiation could be due to the local electric field generated by 2D nanostructured ZnO nanosheets. Although there are several studies describing osteoblast gene expression under electrical stimulation [47,56], there is not a consensus of how gene expression is modulated under electrical stimulation. This is due, in part, to the variability in the parameters used in each study (i.e., voltage, time under electrical stimuli, cell line, etc.), which hampers comparisons between studies. In addition, in the present work, it was not feasible to measure the voltage or times that cells had been stimulated, because this depends on the cell movements. However, the generation of a local electric field due to cell movement, without the need of external stimulation, is an important point for bone tissue regeneration.

## 5. Conclusions

In summary, we can conclude that the hybrid material made of a metallic TiZrPdSiNb alloy and covered with hydrothermally grown ZnO nanosheets is not cytotoxic. Additionally, the material does not induce an inflammatory response (i.e., cytokine release is null or very low for TNF-α, IL-6 or IL-1β) after 24 h in culture and, more importantly, it improves the proliferation of Saos-2 cells by a factor of 1.5 after seven days in culture. Finally, it enhances the expression of some early differentiation expression genes without the need of any electrical or chemical external stimuli. In particular, the levels of the studied genes up to 21 days are similar or higher for cells grown on ZnO nanosheets as compared to the uncoated alloy (except for IBSP on day 14) and ZnO thin film (except for COL1 and IBSP on day 14 and SPARC on day 21). The generally higher levels detected for the ZnO nanosheets are due to a self-stimulation mechanism which is triggered by the motion of cells and the concomitant strains generated with the ZnO nanosheets. As a result, local electric fields which boost cell proliferation and differentiation built up due to piezoelectricity. These beneficial effects were not observed when a flat ZnO thin film was deposited on the alloy surface by sputtering because of the full mechanical clamping. The obtained results will certainly have an important impact on surface nanoengineering of smart bioimplants with enhanced performance.

## Figures and Tables

**Figure 1 biomedicines-09-00352-f001:**
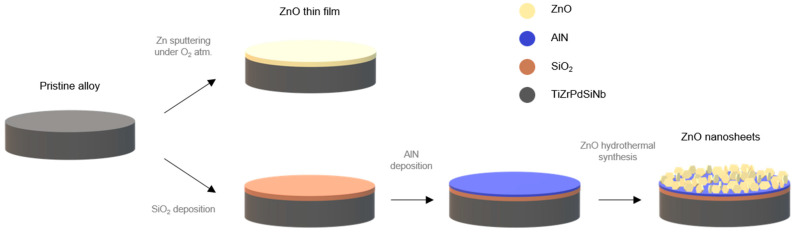
Schematic illustration of the coating process to obtain ZnO thin films (sputtering) and ZnO nanosheets (hydrothermal synthesis) on TiZrPdSiNb alloy.

**Figure 2 biomedicines-09-00352-f002:**
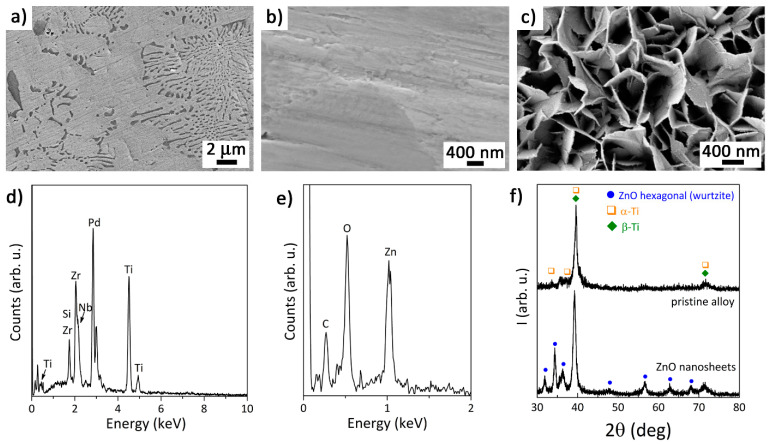
SEM images of (**a**) pristine alloy, (**b**) ZnO thin film, and (**c**) ZnO nanosheets deposited on top. EDX patterns of (**d**) pristine alloy and (**e**) ZnO thin film samples. (**f**) Grazing incidence X-ray diffraction (GIXRD) patterns of pristine alloy and ZnO nanosheets.

**Figure 3 biomedicines-09-00352-f003:**
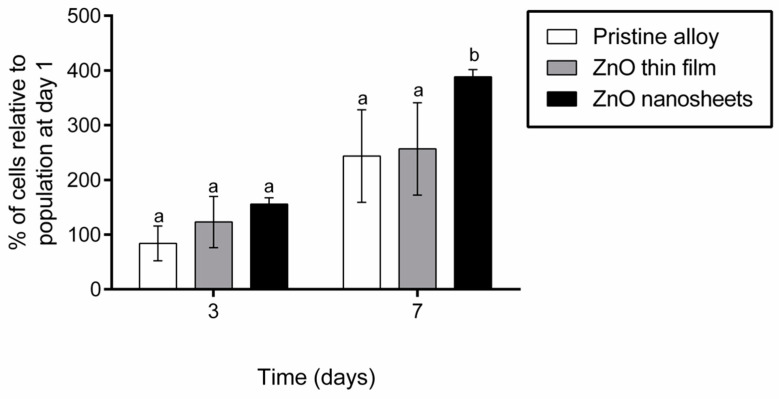
Proliferation of Saos-2 cells grown on pristine alloy, ZnO thin film, and ZnO nanosheets at 3 and 7 days in culture. Results were normalized by day 1. Different superscripts on top of the columns denote significant differences (*p* < 0.05) among the materials at the same time-point.

**Figure 4 biomedicines-09-00352-f004:**
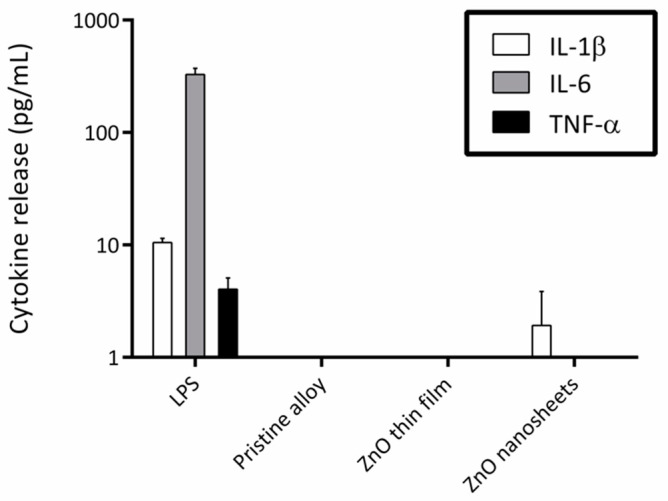
Logarithmic representation of cytokine release by macrophages cultured in the presence of lipopolysaccharide (LPS) (positive control) and the three different samples. Secretion was analyzed by the cytometric bead array (CBA) test at 24 h of culture. Error bars indicate the standard error of the mean of three replicas.

**Figure 5 biomedicines-09-00352-f005:**
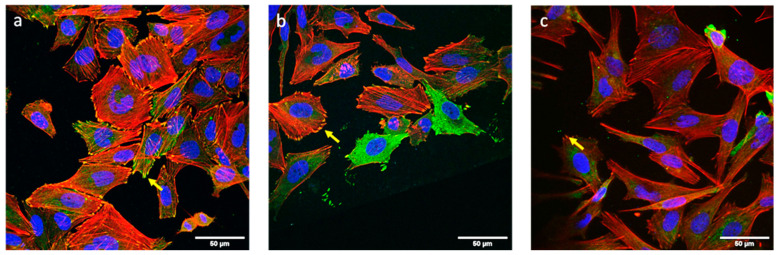
Saos-2 cells adhered to the surface of the pristine alloy (**a**), ZnO thin film (**b**), and ZnO nanosheets (**c**) after 3 days in culture. Stress fibers (actin; red), vinculin (green) and nuclei (DNA; blue) can be observed. Focal contacts (yellow arrows) appear as yellow spots due to the overlay of actin and vinculin signals.

**Figure 6 biomedicines-09-00352-f006:**
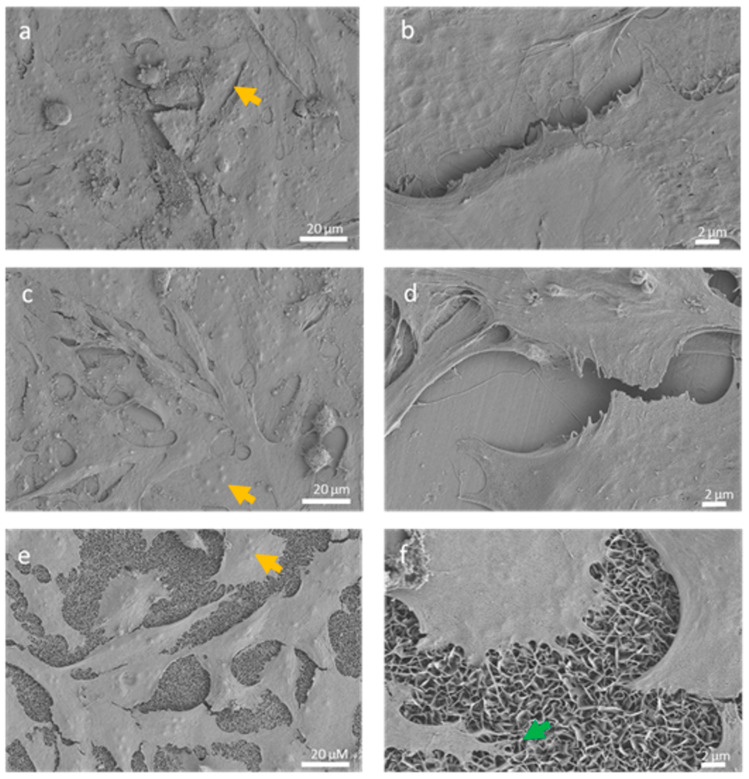
SEM images of Saos-2 cells cultured for 7 days on pristine alloy (**a**,**b**), ZnO thin film (**c**,**d**), or ZnO nanosheets (**e**,**f**). Several nucleoli can be seen inside nuclei (yellow arrows). Long projections are anchored directly on the nanosheets (green arrow).

**Figure 7 biomedicines-09-00352-f007:**
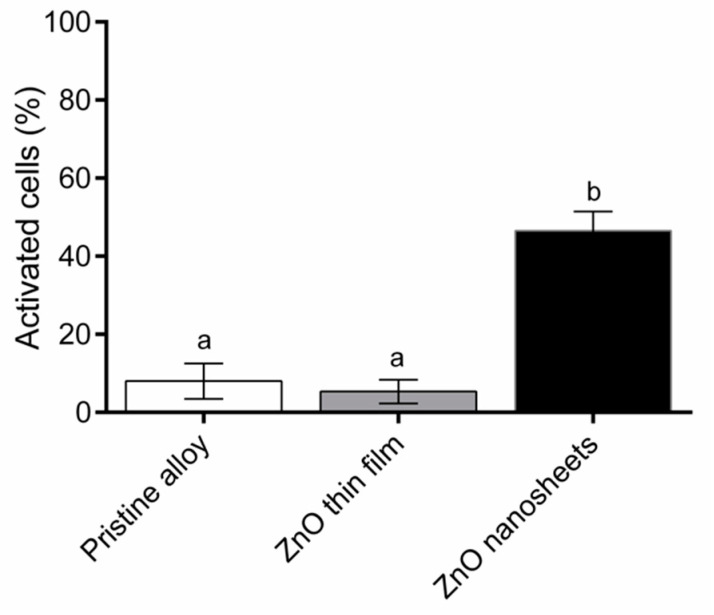
Percentage of activated osteoblastic (Saos-2) cells which underwent transient changes in intracellular calcium concentration when grown on top of the different samples. Different superscripts on top of the columns denote significant differences (*p* < 0.05) among the materials.

**Figure 8 biomedicines-09-00352-f008:**
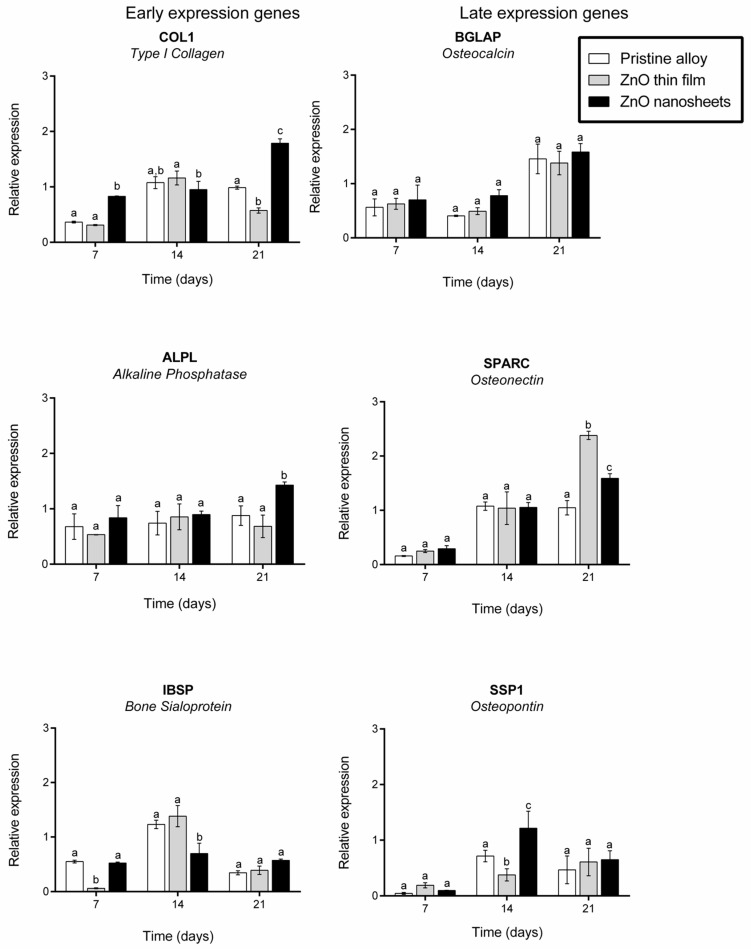
Quantification of mRNA levels. Relative expression of osteoblast differentiation markers COL1, ALP, BGLAP, IBSP, SPARC and SPP1 (under the protein encoded) in Saos-2 cells on day 7, 14 and 21 after seeding on the three samples. The target gene levels are expressed as a relative value. Different superscripts on top of the columns denote significant differences (*p* < 0.05) among the materials at the same time-point.

## Data Availability

The data presented in this study are available on request from the corresponding author. The data are not publicly available due to their use in further studies.

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
