# Peer review of "ZnO Nanosheet-Coated TiZrPdSiNb Alloy as a Piezoelectric Hybrid Material for Self-Stimulating Orthopedic Implants"

_biomedicines, 2021, doi:10.3390/biomedicines9040352_

Round 1
Reviewer 1 Report
I consider this paper is important in the research field related to orthopedic implant. The description of the protocols used is comprehensive, replicable and will serve as a good reference for searchers working in the field.. The statistical analysis the data been performed appropriately.
This manuscript is technically sound and do the data support the conclusions.
I recommend adding more details on the choice of Human osteosarcoma Saos-2 cells for this study.
Author Response
Manuscript ID: biomedicines-1162755
Type of manuscript: Article
Title: ZnO nanosheets-coated TiZrPdSiNb alloy as a piezoelectric hybrid material for self-stimulating orthopedic implants
Authors: Oriol Careta, Jordina Fornell, Eva Pellicer, Elena Ibañez, Andreu Blanquer, Jaume Esteve, Jordi Sort, Gonzalo Murillo, Carme Nogués *
AUTHOR RESPONSES TO REVIEWER 1 COMMENTS
I consider this paper is important in the research field related to orthopedic implant. The description of the protocols used is comprehensive, replicable and will serve as a good reference for searchers working in the field. The statistical analysis the data been performed appropriately. This manuscript is technically sound and do the data support the conclusions.
I recommend adding more details on the choice of Human osteosarcoma Saos-2 cells for this study.
We would like to thank the reviewer for the positive comments.
According to his/her recommendation, we have added a new sentence to explain why the Saos-2 cell line was chosen for the study (line 161).
“Saos-2 cell line derives from osteosarcoma, and even though it could show some minor disparities in terms of cellular behaviour when compared with primary human osteo-blasts, Saos-2 cells have been widely used to test biocompatibility of newly developed materials for a wide range of biomedical applications, being considered as a representative model of multiple osteoblasts responses [ref1].
[ref1] Saldaña, F. Bensiamar, A. Boré, N. Vilaboa, In search of representative models of human bone-forming cells for cytocompatibility studies, Acta Biomater. 7 (2011) 4210–4221. https://doi.org/10.1016/j.actbio.2011.07.019.
Reviewer 2 Report
1. In this part of the article "as they exhibit superior mechanical properties, biocompatibility and corrosion resistance" it is necessary to provide specific links to sources. In addition, indicate that the predictive properties of ternary and multicomponent alloys allow the study of computer modeling (as an example, you can include in references DOI 10.1007/s11665-018-3358-y 10.1016/j.matchemphys.2019.121895,
2. Page 1 "alloys face some problems like the presence of non-desirable elements, such as Al and V in Ti6Al4V alloys, or an exceedingly low stiffness" This statement is highly controversial.
3. In conclusion, I think it makes sense to indicate and give numerical conclusions for the studied properties.
Author Response
Manuscript ID: biomedicines-1162755
Type of manuscript: Article
Title: ZnO nanosheets-coated TiZrPdSiNb alloy as a piezoelectric hybrid material for self-stimulating orthopedic implants
Authors: Oriol Careta, Jordina Fornell, Eva Pellicer, Elena Ibañez, Andreu Blanquer, Jaume Esteve, Jordi Sort, Gonzalo Murillo, Carme Nogués*
AUTHOR RESPONSES TO REVIEWER 2 COMMENTS
First at all, we would like to thank the reviewer for the comments and suggestions.
1.“In this part of the article "as they exhibit superior mechanical properties, biocompatibility and corrosion resistance" it is necessary to provide specific links to sources. In addition, indicate that the predictive properties of ternary and multicomponent alloys allow the study of computer modeling (as an example, you can include in references DOI 10.1007/s11665-018-3358-y 10.1016/j.matchemphys.2019.121895, “
The following references have been added as suggested by the reviewer.
“as they exhibit superior mechanical properties, biocompatibility and corrosion resistance [ref1, ref2], when compared with other…”
[ref1] S. Ghosh, S.M. Dasharath, S.Mula, Simulation Kinetics of Austenitic Phase Transformation in Ti+Nb Stabilized IF and Microalloyed Steels, J. Mater. Eng. Perform. 27 (2018) 2595-2608. https://doi.org/10.1007/s11665-018-3358-y
[ref2] M. Niinomi, Mechanical properties of biomedical titanium alloys, Mat. Sci. Eng: A. 243 (1998) 231-236. https://doi.org/10.1016/S0921-5093(97)00806-X
- Page 1 "alloys face some problems like the presence of non-desirable elements, such as Al and V in Ti6Al4V alloys, or an exceedingly low stiffness" This statement is highly controversial.
We agree with the reviewer that this sentence might be misleading. For clarity, we have rephrased it.
“Nevertheless, these alloys face some problems like the presence of non-desirable elements, such as Al and V in Ti-6Al-4V alloys [ref3, ref4], or the mismatch between their Young’s modulus (i.e. ~ 110 for Ti-6Al-4V [ref2] and 40-80 for Ti-Nb alloys [ref5] and that of the bone (10-30 GPa) [12].
[ref3] U. Kamachimudali, T.M. Sridhar, B. Raj, Corrosion of bio implants. Sadhana, 28 (2003) 601-637. https://doi.org/10.1007/BF02706450
[ref4] S. Karimi, Corrosion behavior of metallic bio-implan alloys, Doctoral dissertation, University of British Columbia, 2014.
[ref5] S. Hanada, H. Matsumoto, S. Watanabe, Mechanical compatibility of titanium implants in hard tissues, Int. Congr. Ser. 1284 (2005) 239-247. https://doi.org/10.1016/j.ics.2005.06.084
- In conclusion, I think it makes sense to indicate and give numerical conclusions for the studied properties
Following reviewer’s suggestion, the conclusions have been partly rewritten to provide quantitative data of the studied properties. The text has been changed as follows:
In summary, we can conclude that the hybrid material made of a metallic TiZrPdSiNb alloy and covered with hydrothermally grown ZnO nanosheets is not cytotoxic. Also, the material does not induce an inflammatory response (i.e., cytokine release is null or very low for TNF-α, IL-6 or IL-1β) after 24 h in culture and, more importantly, it improves proliferation of Saos-2 cells by a factor 1.5 after 7 days in culture. Finally, it enhances the expression of some early differentiation expression genes without the need of any electrical or chemical external stimuli. In particular, the levels of the studied genes up to 21 days are similar or higher for cells grown on ZnO nanosheets as compared to the uncoated alloy (except for IBSP on day 14) and ZnO thin film (except for COL1 and IBSP on day 14 and SPARC on day 21). The generally higher levels detected for the ZnO nanosheets are due to a self-stimulation mechanism which is triggered by the motion of cells and the concomitant strains generated to the ZnO nanosheets. As a result, local electric fields which boost cell proliferation and differentiation built up due to piezoelectricity. These beneficial effects are not observed when a flat ZnO thin film is deposited on the alloy surface by sputtering because of the full mechanical clamping. The obtained results will certainly have an important impact on surface nanoengineering of smart bioimplants with enhanced performance.